# Digitalisation, Parenting, and Children’s Mental Health: What Are the Challenges and Policy Implications?

**DOI:** 10.3390/ijerph19116452

**Published:** 2022-05-26

**Authors:** Jason Hung

**Affiliations:** Department of Sociology, University of Cambridge, Cambridge CB2 1SB, UK; ysh26@cam.ac.uk

**Keywords:** digitalisation, digital literacy, mental health, parenting

## Abstract

In this narrative review, the relationships between digitalisation and the mental health status among children and youths will be discussed first. Second, amidst the pandemic, parents and children have much more time to spend together at home, so parenting plays a more significant role in determining children’s wellbeing. Therefore, how different parenting styles impact digitalisation and the mental health experienced by children and youths will be assessed. This narrative review aims to investigate the parenting conditions needed for children and youths to benefit from the growth of digitalisation, so that policies for digital transformation can be recommended. The outputs of this narrative review include recommending the endeavours of digital transformation that involve indiscriminately active inclusion and facilitating privileged young people, as well as their less advantaged counterparts, to maximise the advantages digitalisation offers.

## 1. Overview

The existing literature argues that the ongoing coronavirus (COVID-19) outbreak significantly impacts children’s emotional, psychological, and social development. For example, children aged three to six years old displayed clinginess and phobia symptoms that their family members might be or become infected with COVID-19; older children aged 6 to 18 experienced inattention in studies [1,2]. Globally, the education, physical activities, and socialisation opportunities of 91% of the students worldwide were at some point disrupted due to the pandemic and the associated need for home confinement [2].

The Council of Europe outlines five areas in relation to social inclusion that European countries should emphasise in order to provide safety nets and prerequisites for the satisfaction of citizens’ needs, including education, the labour market, health, and participation [3]. Since digitalisation has significantly been accelerated due to the outbreak of the pandemic, this narrative review discusses the associations between digital participation in education, the labour market, and mental health for the purpose of understanding how the lack of opportunities for digital participation would pose benefits or detrimental effects on children’s life chances and wellbeing. A supporting statement was presented by the Council of Europe and the European Commission [3], which states, “Digital tools and instruments are either targeted at the formal education system or are aiming to improve young people’s access to employment opportunities (Page 17).” This suggests that digital literacy is increasingly necessary for children to participate in formal education and professional jobs; otherwise they might be digitally and socially left behind, leading to the inadequacy of life-chance acquisition. Unfortunately, an outcome of rapid digital development and the growth of Internet access is that a landscape of inequality caused by varying access to digital resources between and within countries has been seen [3]. This means digital exclusion occurs profoudnly. To combat the entrenched digital inequality and exclusion, highlighting the importance of promoting digital inclusion has significant scholarly value.

The experience that digital societies and economies offer is skewed, relating to the barriers to life chances for young individuals of disadvantaged or marginalised social, economic, and spatial profiles [3]. This shows that unless digital and youth policies are aptly arranged and delivered, digitalisation could compound the disadvantages of digitally inactive or underactive youths, who are predominantly represented in the socioeconomical and sociospatial underprivileged populations.

The outputs of this narrative review include recommending the endeavours of digital transformation that involve indiscriminate active inclusion and facilitating privileged young people, as well as their less advantaged counterparts, to maximise the advantages digitalisation offers. In order to achieve this goal, the Council of Europe and European Commission [3] stresses that youths should be encouraged to become involved in the design and development of digital transformation policies. In doing so, governmental agencies can arrange and deliver a better digital platform that suits the educational, professional, social, and other needs of the youths themselves.

Existing studies demonstrate that digitalisation and home confinement affect children and youths’ mental health. In China, 2330 students from grades two to six at two primary schools in the Hubei province where Wuhan is located were asked to complete an online survey from 28 February 2020, to 5 March 2020 [4]. Xie et al. [4] found that 403 students (22.6%) suffered from depression symptoms, while 337 students (18.9%) suffered from anxiety traits. Despite Xie et al.’s survey failing to identify the causes of the high depression and anxiety rates, Singh et al. [2] argue that the cancellation of in-person academic activities due to home confinement served as a major factor in the students’ encounters with depression and anxiety. In this narrative review, the relationships between digitalisation and the mental health status among children and youths will be discussed first. Second, amidst the pandemic, parents and children have much more time to spend together at home, so parenting plays a more significant role in determining children’s wellbeing. Therefore, how different parenting styles impact digitalisation and the mental health experienced by children and youths will be assessed. This narrative review aims to investigate the parenting conditions needed for children and youths to benefit from the growth of digitalisation, so that policies for digital transformation can be recommended.

## 2. Digitalisation and Mental Health

Ramtanen et al. [5] argue that mental health services are increasingly digitalised, and young people suffering from poor mental health are at greater risks of digital and, therefore, social exclusion. This suggests that mental health cannot be discussed thoroughly without addressing digitalisation, and vice versa. The European Commission’s 2016 expert group on the “risks, opportunities, and implications of digitalisation for youth, youth work, and youth policy” argued that it is important to foster more equitable access to Internet services and to develop young people’s digital skills so as to ensure they can maintain their competitiveness in the employment market [3]. Here, the emphasis of the European Commission implies that during the digital era, it is important to ensure that disadvantaged children and youths can remain academically and professionally involved in order to facilitate their chances for upward social mobility. While enhancing social mobility opportunities does not necessarily improve the mental health of young cohorts, being digitally and academically excluded, as suggested by Singh et al. [2], is detrimental to youngsters’ mental health.

Metherell et al. [6] echoed the above arguments, finding that a lack of technological access necessary to support online interactions would result in negative mental health outcomes. Moreover, they revealed that digital exclusion discourages children and youths from participating in educational activities during home confinement and online education, further exacerbates mental health issues in children. Metherell et al.’s [6] findings strengthen the positive relationship between digital exclusion and mental health challenges. As the world is becoming increasingly digitalised, digitally excluded school-aged youngsters who suffer from mental health issues would be seen as even more vulnerable. One of the primary reasons why digitally excluded school-aged children and youths are more mentally fragile is because they experience a higher degree of educational and social disruption relative to their digitally engaged counterparts. The lack of Internet access and computers preclude ongoing, active participation in online education and connections with peers [6]. Since Metherell et al.’s [6] study focuses exclusively on the British context, it would be interesting to understand further if such a context applies to socially, economically, and digitally less developed societies. Since less developed societies experience significantly less access to digital services, thus digital exclusion may not affect children’s mental health adversely as much as in Britain. This is because digital inequalities may be narrower in less developed societies.

It is noteworthy that some children and youths may be trapped in a vicious cycle of digital exclusion and mental health issues. Not only does digital exclusion jeopardise mental health, but Rametanen et al. [5] unveiled that mental health issues, especially psychosis, propel digital exclusion. Mentally ill individuals, specifically those suffering from psychosis, may face difficulties in recognising digital information. They may not be able to gain access to e-health services without the help of significant others, such as parents, teachers, or social workers. Moreover, globally, individuals needing to seek mental health services may be fearful of stigmatisation, preventing them from reaching out to professionals in a timely fashion [5]. The provision of e-health services may be able to eradicate or minimise the stigma against those reaching out for psychiatric or psychological support because such services can be delivered while the privacy of recipients receiving services is highly secured. Compared to older counterparts, younger individuals with mental health issues are inclined to have more digital skills, familiarity, and confidence to gain access to e-mental health services [5]. Therefore, the provision and massification of e-mental health services can be viewed as a possible instrument to minimise the mental health challenges suffered by children who are more digitally literate.

Metherell et al.’s [6] findings show that mental health issues suffered by students were worst during the times when there was a high degree of COVID-19 restrictions when both social and educational disruptions peaked. However, when schools reopened and social restrictions were lifted, partially or in full, students’ mental health had to some degree, recovered. In the Chinese context, studies from Singh et al. [2] and Peng [7] support that of Metherell et al. [6], arguing that the cancellation of in-person academic and extra-curricular activities due to home confinement worsened students’ mental health significantly. However, these studies overlooked whether mental health challenges are also posed by in-person schooling as well. For example, those children and youths who experience school bullying and academic stress consistently or those who live far from school may find their mental health risk further at stake if they have to return to school. On the contrary, these individuals may find e-learning more pleasant and favourable for their mental health.

In addition, when students increasingly rely on the Internet for daily activities, their risk of being victims of cyberbullying increases. Ye et al.’s [8] findings demonstrate that 26.9% of middle school respondents in China purportedly experienced cyberbullying. Cyberbullying is harmful to children’s academic engagement and mental health in the form of anxiety, depression, and suicide, fatal or not. Among respondents aged 12 to 18, 43.7% of children encountered depression symptoms, with an additional 37.4% of youths suffering from anxiety symptoms [8]. The Council of Europe and European Commission [3] also urges European governments to better protect the identities of online learners. Only by aptly arranging digital youth policies can school-aged students who are digitally active avoid the experience of cyberbullying; otherwise those who are constantly attacked online may put their mental health at stake.

When mental health is jeopardised, Rametanen et al. [5] argued that marginalised young people have a preference for having in-person interactions over web-based interactions with medical professionals and social workers to discuss their personal problems, despite being digitally literate. However, it is noteworthy that offline health services have a significant geographical barrier. For e-health services, individuals can receive distance medical consultations; thus, medical experts do not have to be present locally to provide medical advice and help. Especially for those dwelling in less developed regions, there may be a lack of the availability of psychiatrists, psychologists, and social workers. Therefore, children and youngsters may not be able to seek face-to-face medical support unless they reach out to online services. As individuals with mental health issues are more likely to doubt the quality of web-based health services relative to their mentally healthy counterparts, local policymakers should carry out pilot studies to introduce and promote the benefits of e-mental health services. However, in developing countries or regions, there is a flood of digital misinformation due to the lack of rigorous online surveillance. Therefore, when non-urban young people seek medical support online, they may find web-based mental health support systems less trustworthy and reliable. It is, therefore, additionally necessary for local policymakers, especially those in rural areas, to strengthen their online surveillance and sanction netizens and health specialists or providers who maliciously spread health-related misinformation.

Winther et al. [9] point out that spending a prolonged duration on online activities would lead to detrimental mental health outcomes, including an increased potential to suffer from depression, social disconnection, and reduced physical activities. Singh et al. [2] support this argument and state that a long course of digital and social media exposure may create or deepen the anxiety suffered by children. However, both studies fail to highlight whether the benefits gained from using the Internet can offset the withdrawal from social participation in in-person events.

Home confinement and social distancing restrictions prompt more children and youths to spend extra time on social media. While Singh et al. [3] criticise that the lengthy browsing of social media worsens school-aged young people’s mental health, Winther et al. [9] argue that such a causal relationship is weak. The weak relationship may result from the fact that the benefits of using social media, especially when in-person interactions are significantly cut due to COVID-19 restrictions, offset the harm of using these online platforms. Additional literature highlights that the growth of social media use is significantly associated with higher risks of, fatal or not, suicide and depression. However, the magnitude of such an association is too weak [10]. This raises the question of how much excessive use of social media would pose a detrimental effect on individuals’ well-being. Another longitudinal research study in the Canadian, American, and European contexts demonstrates that the use of digital technologies (defined by social media, texting, and the Internet), and children’s mental health (exclusively measured by anxiety and depression), are not associated [11]. Again, the findings challenge the general perception that the excessive use of digital technologies could inflict more mental health issues on children.

Young et al. [12] further claim that the use of social media would not pose mental health challenges but benefits youngsters. First, Young et al. [12] focus on the general rather than the lengthy use of social media. Second, their study shows how the issue of the use of social media and children’s mental health is associated should vary among contexts. Shaw and Gant [13] indicate that the use of social media substantially reduces the sense of loneliness and depression while boosting children’s self-esteem. This suggests that the use of social media can enhance emotional and mental wellbeing. However, such literature fails to mention whether the frequency and duration of the use of social media are significant determinants of youngsters’ wellbeing. Furthermore, different social media outlets are designed for varied purposes. For example, LinkedIn is used to circulate achievements and build virtual professional connections, while Instagram is a platform for social sharing. Using different social media may affect children’s well-being differently. There is no single assertive answer to confirm whether the use of social media benefits or harms young people’s wellbeing.

Further, for the use of digital technologies, a stable and fast Internet connection is conducive to children’s mental health [6]. Here a satisfactory Internet connection speed is particularly required for gaming rather than studying. For students who are obsessed with online games, any experience of an undesirable Internet connection speed could be frustrating, plausibly worsening gamers’ mental health. Additionally, Metherell et al. [6] reiterate that for those who face digital exclusion, their encounters with educational disruptions are much more severe relative to their digitally included counterparts. Future research should thoroughly investigate how Internet use affects the mental health of children because school closure, home confinement, public health threats, and other factors all, more or less, appear to play a role in affecting children’s wellbeing. Since there are ample factors of children’s encounters with mental health challenges, researchers have to keep as many relevant external environmental factors constant when investigating whether the use of Internet or social media causes children to suffer from mental health issues.

## 3. Parenting

In this section, a range of major parenting styles, namely authoritative parenting, authoritarian parenting, and indulgent/democratic parenting, are introduced. The author discusses how the relationships between parenting styles and development of children’s mental health are nuanced and vary according to contexts.

Childhood can be considered a developmental period of psychosocial vulnerability [13,14]. Children, compared to adults, engage in more risky behaviours, value danger less, and seek more novelty and stimulation [14,15]. Some problems may be the consumption of alcohol [13] and other drugs [16], lower performance in school [17], challenges to self-concept [18] and self-esteem [19], or difficulties forming healthy attachmentments to equals and parents [20]. However, there are important differences among children in the levels of competence and adjustment, differences that are partly explained by the main contexts in which children develop, such as family [21], school [22], or peers [23].

Parents are a major source of protection but also of risk for the psychosocial health of children [24,25]. Traditionally, the orthodox, so-called authoritative style (based on rigour and warmth) has been identified as the best strategy to promote psychosocial wellbeing [14,26,27]. However, this does not always correspond to the literature. The cultural context in which parents raise their children has been proposed as an important factor in explaining the discrepant results on which is the best parenting method [28,29,30]. On the one hand, studies on ethnic minorities in the United States, such as African-Americans [31] and Chinese-American [28], as well as studies in Arab societies [32], identify the developmental benefits associated with rigour, but without warmth (the so-called authoritarian style). On the other hand, a growing body of studies, mainly conducted in European and Latin American countries, identify the benefits associated with a parenting style based on warmth but without control (the so-called indulgent or democratic style) that offers extensive benefits to the children of these families [19,33]. Children from families with warmth, but without rigour, in comparison with their peers from families with warmth and rigour (authoritative), report equal or even higher indices of wellbeing and competence [19,26]. Likewise, emerging studies analyse the consequences of parental socialisation beyond the years of socialisation [34,35,36]. There is a time when parental socialisation has ended: when the child reaches adult age [19,34]. However, their competence or adjustment is also related to the type of family in which they were socialised [34,37]. Patterns of parental affection and strictness during the socialisation years, as in studies of children, appear to have a theoretically consistent relationship with the adjustment and competence of adult children, including young adults [38], middle-aged adults [19], and older adults [34].

Parents are children’s explicit important significant others, and a healthy parent–child relationship may plausibly be conducive to a positive development of children’s mental health. When constructing positive parenting—parenting that generates positive outcomes in children’s development—and family relationships under the influence of digital technology, especially where authoritative parenting and democratic parenting apply, parents need to employ “positive digital media role modelling.” This means parents should act as their children’s role models and avoid the addictive use of digital technologies. They could also co-explore online content with their children, in addition to supporting children’s safe, appropriate technological use [39]. When children experience negative emotional and psychological well-being due to the prolonged use of social media and the Internet or lengthy home confinement, their parents should also teach them skills to cope with mental health challenges [2].

Parents should listen to and communicate with their children regularly to understand their well-being. If their children encounter difficulties in coping with stress or negative emotions, parents could offer advice on relevant problem solving [2]. Walper and Kreyenfeld [40] support the claim that active communication initiated by both parents and children is necessary to further enhance the quality of parenting, leading to a higher chance of practising positive parenting. It is essential to understand that parent–child communications should not be restricted to talking and giving advice but listening as well. Parents who patiently listen to their children’s encounters with emotional, psychological, or other challenges can foster the willingness to share their struggles. By applying proactive listening and exchanging ideas, parents can serve as reliable figures that their children can rely on whenever they face any unsatisfactory internal or external conflict. With constructive parental support, even if children, for whatever reasons, suffer from mental health issues, they are likely to weather the storm of adversity. However, in the context where authoritarian parenting optimises children’s wellbeing, parents’ proactive listening and communications with children may not be conducive to the mental health of children themselves.

It is especially crucial to discuss parenting styles amidst the pandemic. This is because students could be on the brink of suffering from extensive mental health challenges, and parental support is necessary to mitigate their children’s emotional and psychological challenges. Such authoritarian parenting is similar to “helicopter parenting;” parents overly monitor their children’s activities and arrange or manage their children’s affairs, even when children are entering their late adolescence [40]. Amidst the pandemic and the epoch of digitalisation, children are increasingly asked to stay home. If helicopter parenting is strictly applied in certain contexts, parent–child conflicts may intensify, worsening a child’s mental health. Additionally, due to school closures and work-from-home policies during the pandemic, parents and children have far more time to spend together relative to the pre-pandemic situation. Applying appropriate parenting is seen as unprecedentedly important, as such a parental approach tremendously impacts youths’ mental health and academic well-being.

While positive parenting can facilitate youths’ academic, emotional, and psychological wellbeing, Peng [7] argues that parenting styles in China are gendered. In traditional domestic settings, fathers are seen as authority figures that economically support and morally guide their children, whereas mothers serve as caregivers. Despite the traditional gendered roles in contemporary urban China, for example, middle-class Chinese parents, including some fathers, perform multiple roles as playmates, teachers, friends, and counsellors in everyday childcare [7]. Compared to more traditional Chinese fathers, their urban, privileged parental counterparts are more willing to take part in the emotional and physical care of children. However, urban, middle-class Chinese mothers, rather than fathers, remain the primary caregivers in domestic settings by far, even if they have full-time jobs and economically contribute to their families [7]. An American study echoes these arguments where fathers are increasingly assuming more parenting responsibilities, yet mothers still assume the majority of parenting duties [7]. As societies are becoming increasingly digitalised, gendered parenting roles may be subject to change, where fathers and mothers, if needed, should adjust their parenting styles to accommodate the academic, emotional, or psychological needs of their children.

Digitalisation significantly impacts parenting, where parents regularly incorporate digital platforms and media into everyday childcare. Parents explore childcare information online, maintain digital communications with their children, and check their children’s academic duties in the online school system [7]. This demonstrates that parenting and digitalisation are significantly tied together, prompting the need to investigate whether today’s gendered parenting should be adjusted to better fit the digital influence and improve the mental health of children.

Peng [7] notes that Chinese mothers spent an average of 4.8 h daily on digital media and technology (including entertainment, work, and childcare), and an average of 1.4 h per day were used for parenting. Alternatively, Chinese fathers spent 4.9 h each day on average using digital media and technology, and only one hour every day on average contributed to parenting. Further findings indicate that urban Chinese mothers were inclined to use digital media and technology to explore online parenting information. This supports the bias in the perception of gendered roles that mothers are natural caregivers while fathers mainly act as breadwinners.

The digitalisation of children’s education has raised parental digital labour in online communications with teachers. However, such digital labour has remained profoundly gendered. Peng’s [7] study shows that mothers were usually responsible for engaging in online communications with teachers, joining and participating in parent–teacher WeChat or QQ groups (i.e., the Chinese digital platforms as alternatives to WhatsApp and Twitter, respectively), adding teachers as virtual friends on social media, and subscribing to schools’ public blogs, among other responsibilities [7]. Fathers usually believed that they did not need to share the digital labour in parenting insofar as they observed that their wives assumed such responsibilities [7]. Fathers who demonstrated impatience in their involvement in digital labour in parenting would either remain silent or neglectful when their wives were communicating with teachers via the use of online platforms or would even criticise their children or their wives as a sign of their lack of patience and interest in parenting [7]. These findings suggest that fathers are inclined to shrink from their parenting responsibilities, as they generally perceive parenting as a maternal duty and their own primary role as one of financially supporting the household. Here, Peng fails to argue if semi-parental absence (where fathers withdraw from parenting) is detrimental to children’s mental health, even if there is a constructive mother–child relationship. Moreover, Peng does not suggest whether fathers’ lack of proactive parenting would induce conflicts between fathers and children, especially when children are dissatisfied with the fact that their fathers are not responsive to taking care of their needs.

Parental involvement in children’s upbringing is crucial, especially amidst the pandemic. Both mothers and fathers should spend more time engaging in indoor activities with their children to help their children avoid misusing digital devices and online platforms and potentially developing a digital addiction or experiencing an exacerbation of anxiety symptoms due to digital addiction. Parents should also regularly communicate with children to help monitor their emotional and psychological wellbeing [41]. Paternal disengagement, especially if mothers work full-time and cannot spend sufficient time providing childcare, can cause discouragement in their children and foster the development of mental health issues amid the digitalised era and public health crisis. As children now spend more time in domestic settings, both mothers and fathers should assume more responsibilities to ensure that the everyday emotional and psychological needs of their children are being met.

One of the factors propelling mothers to engage in full-time work is due to the circumstance of single parenting. For single-parent families, not only do children face parental disengagement in parenting, but also such households are often subject to financial constraints. In Australia, Maalsen and Gurran [42] argue that a growing number of single mothers have to use digital accommodation services, such as Airbnb, to find tenants to share their accommodations in order to lessen their own financial burdens. Using digital accommodation services to share accommodations with strangers minimises parents’ ability to fully claim the agency over a household space and the exercise of parenting without any form of interference. Children’s mental health may be at risk if tenants have interpersonal conflicts with them or their single parents. Aside from sharing accommodations that could potentially lead to conflicts, there is a lack, if not an absence, of parenting among single-parent families. Walper and Kreyenfeld’s [40] findings showed that 52% of single mothers or fathers indicated that they did not have sufficient time to take care of their children as they had to also assume responsibilities as breadwinners. If children realise the financial pressures of their single parents, they might endure a higher degree of mental health burdens and academic stress. These could either motivate them to study harder in order to seek better living standards for their families in the long-term or harm their wellbeing due to the unbearable amount of stress.

## 4. Conclusions and Policy Implications

One of the advantages shared by young communities is that they are more adaptive to transformation, digital or otherwise. Therefore, they are more likely, compared to older counterparts, to accept and learn new technologies. Individual governments should therefore market and promote the benefits of developing digital literacy. They should also subsidise poorer cohorts to study courses in relation to digital skills. In the labour market, governments should subsidise companies to provide professional training in digital learning to all employees as well. In doing so, individual governments can ensure more people, especially younger people, have developed sufficient digital skills and knowledge to take advantage of the digital era, including using e-mental health services. For youngsters who suffer from severe mental health challenges, they might, however, be unable to recognise and process digital information. Parents, teachers, and social workers should take the initiative to help these vulnerable populations gain access to e-mental health services. When e-mental health services are widely applied and delivered, those of rural, economically poorer origins can also seek mental health support online, as digitalisation erases the geographical barriers to medical consultation and the delivery of any therapies. Additionally, seeking e-mental health services can minimise the exposure of individuals’ personal identities compared to those looking for offline services. Therefore, mentally ill individuals can worry less about social stigmatisation when they reach out for e-mental health services.

A major concern for digital transformation is seen when individuals endeavour to build digital learning and medical environments. There could be a raft of misinformation, leading to the public’s loss of trust in online details. Individual governments and regional organisations, such as the European Union and the Association of Southeast Asian Nations, should set up agencies to monitor digital order and exploit online surveillance. Those who disseminate misinformation should be fined or even detained. It is important for governments and organisations to deliver the message that netizens should be held accountable when they behave wrongfully online. Only by rigorously monitoring netizens’ behaviours and ensuring that online users understand the importance of self-discipline in any digital platform can children build up the confidence to trust online medical information when they attempt to search for mental health advice.

Moreover, parents, from both maternal and paternal sides, should simultaneously adjust and adapt to accommodate the needs of their children. For those who have sufficient digital literacy, both mothers and fathers should proactively utilise digital media and technology to better engage in everyday parenting-related activities. They should also spend more family time with their children at home to strengthen the parent–child relationship and ensure their children can achieve satisfactory emotional and social development. Parental presence, engagement, and communications with children regularly, alongside parental respect for children, are the cornerstones to ensure that school-aged children can optimise their mental health despite the ongoing public health threats and home confinement rules.

The importance of parenting is restricted to not only children’s mental health development but also the well-being of adult children once parental socialisation is over [43]. Challenges for parents with children in the digital society are no longer limited to preventing problems inflicted on their children, such as the use of drugs or premature, unsafe sex [17], but also promoting a good self-concept alongside health and well-being among the children [24]. It is, last but not least, noteworthy that most parental socialisation studies have methodological flaws, such as the limited sample sizes. Therefore, large sample sizes should be addressed in future, relevant research to result in statistically just outputs that examine the nuanced relationships between parenting and children’s mental health amid the digital era [44].

## Data Availability

Not applicable.

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
