# Peer review of "Digitalisation, Parenting, and Children’s Mental Health: What Are the Challenges and Policy Implications?"

_ijerph, 2022, doi:10.3390/ijerph19116452_

Round 1

Reviewer 1 Report

The present manuscript explores the challenges of parental socialization in the context of digital society. A detailed comprehensive review is presented considering the importance of family, culture and digital tools (e.g., QQ chat). Parents perform a general socializing task to promote children's development. But, at the same time, parents and children are in a digital society where technologies are very important, so parents must help and teach their children to face technological challenges. 

Some points should be corrected:

- The manuscript seems to be the so-called narrative review. It should be used the term “narrative review” instead of “essay” within the manuscript.

- Authors use the statement of the European Commission “Digital tools and instruments are either targeted at the formal education system or are aiming to improve young people’s access to employment opportunities”. If it is possible, authors should use for literal quotation the number of the page or pages.

-Since this is an international magazine with readers from all over the world, the authors should explain very briefly what WeChat and QQ, which are frequently used in China but less well known in other countries.
- “Parental involvement in children’s upbringing is crucial especially amidst the pandemic. Both mothers and fathers should spend more time engaging in indoor activities with their children to help their children avoid misusing digital devices and online platforms and potentially developing a digital addiction, or experience” (see the paragraph of the pages 9 and 10). This paragraph should have at least one or a couple of references which support the statements.

- The references should be reviewed in detail because the same reference appears with different numbers. Each reference should have a unique number, if the same reference is cited in different parts of the text, the same number should be used. For example, in this reference:

Council of Europe and European Commission. Young People, Social Inclusion and Digitalisation: Emerging Knowledge for Practice and Policy. Paris: Council of Europe Publishing 2021. 

It appears as reference number 4,5 6, 7, 13 and 28, when is the same reference, nor different.

The mistakes in references should be addressed and carefully checked.

Author Response

  1. In response to the reviewer's comment, I already changed the word "essay" to "narrative review" throughout the paper.
  2. I added the page number when directly quoting a sentence from the literature.
  3. I briefly clarified what WeChat and QQ are.
  4. I added a citation in accordance with the reviewer's comment.
  5. References are reorganised and no duplicated references are seen now.

Reviewer 2 Report

The article addresses digitalization, parenting, and mental health, which are interconnected topics of great importance to public health. Overall, the article is well written, concisely structured, and easy to read. However, there is an important problem with the references, as several of them, as I describe below, are repeated. I suggest that the authors solve this problem that, for me, makes publication unfeasible.

REPEATED REFERENCES:

Singh - 2,3,11,14,23,31,32,81,82

Council – 4-8,13,28

Xie – 9,10

Metherell – 15-18,22,25,39,40

Ramtanen – 12,19-21,29

Peng – 24,85-87,89-94

Ye – 26,27

Winther – 30,33

Young – 36,37

Fuentes – 41,45

Queiroz – 55,69

Steinberg – 42,43

Darling – 54,60

Gimenez-Serrano – 70,79

Martinez-Escudero – 74,78

Walper – 83,84,88,96

Peng – 85-87,89-94

Author Response

1. In response to the reviewer's comments, I already reorganised the reference list. Now no duplicated references are seen in the attached, revised manuscript. Thanks!

Reviewer 3 Report

The essay presents an interesting overview about digitalization, mental health, and parenting.The essay is interesting and nice to read, and presents the concept with a good flow. I have some minor concerns, detailed below.

  • There seems to be a lack of coherence in the manuscript about the age of the population under investigation, especially when it comes to the taxonomy employed. For example, on Line 24 individuals aged up to 18 are called older children, while later on in the manuscript they are referred to —more correctly in my opinion— teenagers. The terminology employed makes a little difficult to understand what the different studies are referring to, in terms of age of the population. I would suggest the author to review the manuscript and be consistent with the terminology employed.
  • There is a major issue with the references. The reference list counts 96 works, but most of the entries are repeated. E.g. Ramtanen et al (2021) appears as 12, 19, 20, 21, 29, Pen et al. 2022 appears as 85-87 as well as 89-94. Please edit the manuscript as such to have each paper only appearing once in the reference list.
  •  On the topic of the references, some appears in the wrong position, while some strong sentence are missing a proper source (e.g. Line 406-408). For example, the reference Xie et al [9] on Line 68 should be introduced first around line 65, where the study is first described.
  • Coming to parenting, I think a proper introduction of the different parenting styles is missing, with just the authoritative parenting style mentioned. As such, is difficult to understand what the author refers to as "positive parenting", which is a term not generally used in parenting studies. I would suggest to edit the section providing a brief overview of the other parenting styles and by using a more appropriate terminology, in line with the literature on the topic.

Minor issues:

  • Line 349, I think it's not the gender perception, but a bias in the perception of genders' roles. I would suggest to rephrase the sentence to make it easier to understand.

Overall, the paper is good and a pleasure to read. I think that with the right amount of edits the paper could be an interesting work worth publishing.

Author Response

  1. In response to the reviewer's comments, I standardised all wordings about "children and adolescents" or "adolescents" to "children." As defined in the manuscript, individuals up to the age of 18 are called children. Therefore, it makes logical sense to address adolescents as children.
  2. I reorganised the reference list. Now no duplicated references are seen in the reference list.
  3. I re-positioned the citations, when necessary.
  4. In the "parenting" section, I added a brief introductory paragraph at the beginning of the section.
  5. I changed the wordings of "gender perception" to "biases in the perception of gendered roles," as requested. 

Reviewer 4 Report

This essay aims to investigate the parenting conditions needed for children and youths to benefit from the growth of digitalization and mental health. The literature review is comprehensive and adequate. It includes the most important research on parenting conducted to date, a very relevant topic in the present essay. The present study also talks about how digitalization and the COVID-19 pandemic affect children and youths’ mental health and how parenting could influence on this relationship.

Some conclusions should be highlighted a little bit more. Authors should highlight a little bit more as one of the conclusions the importance of parents not only for children and adolescents, but also for adult children once the parental socialization is over. In this sense, a recent study related parental warmth and strictness with adjustment in adult children from three family generations (Garcia et al., 2020). It should also be noted that the challenges for parents with adolescent children in the digital society are no longer limited to preventing problems for their children, such as drug use (Hernandez-Serrano et al., 2021 ), but also to promote good self-concept as well as health and well-being (Sacca et al., 2021). Finally, another conclusion should focus on the methodological limitations of many parental socialization studies, which should include larger sample sizes to achieve an adequate statistical power (Perez et al., 1999).

The content of Reference section is correct but there are some typographical errors in this section. There are some references repeated. For example, references 4 to 8 are exactly the same. The same is true for references 9 and 10 and for other references more. Authors should review the entire References section to solve it.

References

Garcia, F., Martínez, I., Balluerka, N., Cruise, E., Garcia, O. F., & Serra, E. (2018). Validation of the five-factor self-concept questionnaire AF5 in Brazil: Testing factor structure and measurement invariance across language (Brazilian and Spanish), gender, and age. Frontiers in Psychology, 9, Article 2250. https://doi.org/10.3389/fpsyg.2018.02250

Garcia, O. F., Fuentes, M. C., Gracia, E., Serra, E., & Garcia, F. (2020). Parenting warmth and strictness across three generations: Parenting styles and psychosocial adjustment. International Journal of Environmental Research and Public Health, 17(20), 7487. https://doi.org/10.3390/ijerph17207487

Hernández-Serrano, O., Gras, M. E., Gacto, M., Brugarola, A., & Font-Mayolas, S. (2021). Family climate and intention to use cannabis as predictors of cannabis use and cannabis-related problems among young university students. International Journal of Environmental Research and Public Health, 18(9308), 1-15.   doi:10.3390/ijerph18179308

Pérez, J. F. G., Navarro, D. F., & Llobell, J. P. (1999). Statistical power of Solomon design. Psicothema, 11, 431-436.

Sacca, L., Rushing, S. C., Markham, C., Shegog, R., Peskin, M., Hernandez, B., Gaston, A., Singer, M., Trevino, N., Correa, C. C., Jessen, C., Williamson, J., & Thomas, J. (2021). Assessment of the reach, usability, and perceived impact of "talking is power": A parental sexual health text-messaging service and web-based resource to empower sensitive conversations with American Indian and Alaska native teens. International Journal of Environmental Research and Public Health, 18(9126), 1-15.   doi:10.3390/ijerph18179126

Author Response

  1. In response to the reviewer's comments, I already added a paragraph at the end of the manuscript as an additional conclusion.
  2. The reference list is reorganised. Now no duplicated references are seen from the list.

This manuscript is a resubmission of an earlier submission. The following is a list of the peer review reports and author responses from that submission.